# Construction Artifacts in Metaphor Identification Datasets

**Joanne Boisson**[1] and **Luis Espinosa-Anke**[1,2] and **Jose Camacho-Collados**[1]

[1]Cardiff NLP, School of Computer Science and Informatics
Cardiff University, United Kingdom
[2]AMPLYFI, United Kingdom
{boissonjc,espinosa-ankel,camachocolladosj}@cardiff.ac.uk

## Abstract

Metaphor identification aims at understanding whether a given expression is used figuratively in context. However, in this paper we show how existing metaphor identification datasets can be gamed by fully ignoring the potential metaphorical expression or the context in which it occurs. We test this hypothesis in a variety of datasets and settings, and show that metaphor identification systems based on language models without complete information can be competitive with those using the full context. This is due to the construction procedures to build such datasets, which introduce unwanted biases for positive and negative classes. Finally, we test the same hypothesis on datasets that are carefully sampled from natural corpora and where this bias is not present, making these datasets more challenging and reliable.

## 1 Introduction

The automatic identification of metaphors in corpora is an active area of research in NLP (Tsvetkov et al., 2014; Shutova et al., 2016; Rei et al., 2017; Wu et al., 2018; Gao et al., 2018; Wang et al., 2023), and consequently, previous works have proposed the construction, curation and testing of metaphor indentification datasets in various forms (Birke and Sarkar, 2006; Tsvetkov et al., 2014), also stemming from previous psycholinguistic research (Cardillo et al., 2017; Jankowiak, 2020). A common feature of metaphor identification tasks is that they are usually framed as binary classification, where a potentially metaphorical expression (PME) and a context are provided as input, and a system has to determine whether the given PME is used metaphorically or not. For instance, *dark* is used metaphorically in the sentence *'The latest developments move us closer to a dark age'*, but not in *'I like dark colors'*. While this setting is attractive for testing supervised systems, it also simplifies the task, introducing a real risk of not testing metaphoricity, but instead spurious correlations that might lead to an artificially correct solutions.

Analyses of this sort have often proven critical in understanding the relationship between a proposed task and whether performance on associated datasets can be directly linked to performance on the task itself. Notable examples include the work of Levy et al. (2015), who showed that, in the task of lexical relation modeling, the unreasonably high performance of supervised systems could be attributed to *lexical memorization*, that is, the presence of large number of prototypical cases in the dataset (e.g. *animal* for the hypernymy relation) that made it simple for models to "detect the relation" without having to consider both words in the pair. This issue has been identified also in word analogies (Linzen, 2016; Drozd et al., 2016; Nissim et al., 2020) and, beyond lexical semantics, in natural language inference (NLI). In NLI, given a hypothesis and a premise, a system must determine whether the premise entails, contradicts or is neutral with respect to the hypothesis. Previous works have shown that supervised models could rely on superficial factors, e.g., in the SNLI dataset (Bowman et al., 2015), hypothesis-only models are surprisingly competitive (Poliak et al., 2018; Gururangan et al., 2018), a trend also observed in medical NLI datasets (Alghanmi et al., 2021).

In this paper, we report experimental results which suggest that (1) language models can identify metaphorical expressions with great accuracy without even *seeing* the given expression; and that (2) a model only seeing the metaphorical expression with no context performs very competitively, in both cases close to the model with complete information. A crucial distinction in this phenomenon, however, is that this is only observed in datasets not sampled from a natural distribution, and that whenever such distribution is preserved, these shortcuts are not as effective. We thus propose a simple sampling procedure from a naturally-distributed corpus

that mitigates this issue.

## 2 Metaphor Identification Datasets

This section presents the list of existing English-language metaphor identification datasets used in our experiments. All the datasets are summarized in Table 1.

### 2.1 Psycholinguistic datasets

Datasets created for Psycholinguistics and Cognitive Science served experiments on the perception of metaphors. They are composed of sentences written or rephrased for the purpose of controlling characteristics such as length and word frequency. This line of study for the English language started with Katz et al. (1988)[1], and was pursued by Cardillo et al. (2010) and Cardillo et al. (2017), who released a corpus of verbal and nominal PMEs (CARD_N and CARD_V). Jankowiak (2020) extended this work for sentences of the form *A is-a B*.

### 2.2 NLP datasets

**Sentence context.** The TroFi dataset published by Birke and Sarkar (2006) has often been used to evaluate metaphor identification NLP systems, Turney et al. (2011) being a notable example. The collection started with 50 predefined labeled target verbs. Then, sentences containing one of those verbs were extracted from the Wall Street Journal (WSJ) corpus, and subsequently labeled. Tsvetkov et al. (2014) used TroFi to train a classifier for verbal metaphor identification that integrates features from WordNet (Fellbaum, 1998). As part of their research, they released a balanced test set of 222 sentences with verbal PME, and a new corpus of adjective-noun pairs (TSV_V and TSV_AN). Only clear examples with strong inter-annotator agreements were included in the test set. Mohammad et al. (2016) studied the correlation between metaphors and emotions. This was achieved by building a new corpus of verbal metaphors from WordNet glosses of polysemous predicates, enriched with synset annotations and emotionality (MOH). Mohler et al. (2016) released the large LLC corpus of metaphors covering several part-of-speech and longer metaphoric expressions, scored on metaphoricity and emotions scales, and enriched

---

[1]This dataset is not included in our experiments because it only contains metaphoric instances.

| Data | # total | %met. | # train | # dev | # test |
|---|---|---|---|---|---|
| **NLP** | | | | | |
| TroFi | 3737 | 57 | 1772 | 0 | 1965 |
| TSV_AN | 1963 | 50 | 1763 | 0 | 200 |
| TSV_V | 3959 | 57 | 3737 | 0 | 222 |
| GUT | 8591 | 54 | | - | |
| MOH | 1632 | 25 | | - | |
| LLC | 7343 | 41 | | - | |
| CHAK | 468 | 67 | | - | |
| DUNN | 60 | 67 | | - | |
| NEU | 100 | 56 | | - | |
| **PIE** | | | | | |
| IDIX | 5519 | 48 | | - | |
| PVC | 1348 | 65 | | - | |
| VNC | 2568 | 79 | | - | |
| SE2013_ALL | 1969 | 60 | 1111 | 341 | 517 |
| SE2013_LEX | 2371 | 51 | 1421 | 357 | 593 |
| MAD | 4558 | 48 | 3609 | 466 | 483 |
| PIE | 3025 | 47 | 786 | 1112 | 1127 |
| MAGPIE_R | 48395 | 75 | 38715 | 4840 | 4840 |
| MAGPIE_L | 48395 | 75 | 38716 | 4839 | 4840 |
| **Psycholinguistics** | | | | | |
| CARD_N | 512 | 50 | | - | |
| CARD_V | 280 | 50 | | - | |
| JANK | 360 | 33 | | - | |
| **VUAC** | | | | | |
| VUAC_DO | 14820 | 51 | | - | |
| VUAC_ST1 | 23113 | 28 | 17240 | 0 | 5873 |
| VUAC_ST2 | 94807 | 16 | 72611 | 0 | 22196 |
| VUAC_BO | 39223 | 52 | | - | |

Table 1: Statistics of the datasets. The original split is included in the table when is was provided by the authors. *#total* shows the total number of instances, and *%met.* indicates the percentage of instances labeled as metaphors.

with source domain information. The dataset construction relies partly on automatic metaphor extraction tools, with a manual validation of the output, resulting inevitably in a bias of the considered instances. Dunn (2014) created a small test set of 60 instances to evaluate a metaphoricity score model (DUNN), with the same verb appearing with different levels of metaphoricity in three sentences sets. Similarly, Chakrabarty et al. (2021) released a small corpus of verbal PME to evaluate the MERMAID model (CHAK).

**Adjective-nouns.** Assaf et al. (2013) developed sets of labeled adjective-noun pairs of concrete-abstract associations (NEU) and Gutiérrez et al. (2016) released a much larger dataset of frequent adjective noun pairs for a study on the compositional properties of metaphors (GUT).

### 2.3 Potential Idiomatic Expressions (PIEs)

Idioms, such as *to rock the boat*, are multiword metaphoric expressions that became lexicalized in

a language. Similarly to the corpus construction methodology used for TroFi, idiomatic expressions datasets are usually built from an initial list of PIEs, which are then used to extract sentences from corpora, mostly from the British National Corpus. Several datasets focusing on different types of PIE have been released: Cook et al. (2008) for verb noun constructs (VNC), Sporleder et al. (2010) for various multiword expressions (IDIX), Tu and Roth (2012) on prepositional verb constructs (PVC), Korkontzelos et al. (2013) in two different tracks (SE2013), and Tayyar Madabushi et al. (2022) for nominal compounds (MAD). Haagsma et al. (2020) released the large MAGPIE dataset containing PIEs of various syntactic constructs, following an initial release of a smaller PIE corpus.

## 2.4 Sampled from annotated corpora: The case of the Amsterdam corpus

The VU Amsterdam Corpus (Steen, 2010, VUAC), is a collection of documents from the British National Corpus (BNC-baby), labeled following the metaphor identification MIPVU protocol. Each word of the documents was considered by the annotators and marked when metaphoric. The dataset covers over 190,000 lexical units. Because very conventional metaphors were labeled as figurative, several versions of the corpus have later been released containing metaphoricity or novelty scores. Other modifications have been made to frame the corpus into NLP system inputs for figurative language identification. For example, Do Dinh et al. (2018) enriched the data with novelty scores (DO) and Leong et al. (2020) released two versions of VUAC for a shared task of metaphor identification, with different sets of Part-of-Speech (PoS) tags considered (ST1 for verbs, ST2 for all PoS tags). The datasets are designed for classification of single tokens in the context of a sentence. Parde and Nielsen (2018a; 2018b) also worked on metaphor novelty in VUAC, releasing a version of potential metaphoric source-target pairs among the syntactic dependencies of a metaphoric word.

**Our sampling method.** Framing the VUAC for binary classification requires to sample literal instances of PME from the corpus. Any token that is not labeled as metaphoric in the VUAC can be considered literal. To avoid PME sampling biases, we sample literal instances from the set of expressions that also occur as metaphors, whenever possible. When the same word sequence is not found, we

rely on identical lemma sequences, and finally on identical PoS sequences (we referred to this VUAC sampling as VUAC_BO).

## 3 Evaluation

The evaluation is aimed at understanding to what extent metaphor identification datasets are affected by construction or sampling biases.

### 3.1 Experimental setting

**Data.** For the initial experiments (Section 3.2), we rely on the original data splits of existing metaphor identification datasets (see Table 1).[2] In Section 3.3 we also provide an extended analysis to assess the impact of baselines in different data splits with 5-folds cross-validation.

**Model and training.** As our model for all the experiments, we rely on BERT-base (Devlin et al., 2019). Note that the goal of the experiments is not to provide the best possible model, but rather to show how a supervised model with incomplete information can attain a performance similar to the model with all the information. We use the HuggingFace transformers library and models, adding a classification layer on top of the pre-trained BERT model. For hyperparameter optimisation, we rely on the Bayesian Optimization with Hyperband (BOHB) algorithm (Falkner et al., 2018) with 50 trials, available in RayTune (Liaw et al., 2018). The hyperparameters search space is set to a batch size equal to 4, 8 or 16; the learning rate in a ranging from 5e-7 to 5e-5; the number of epochs within 1 to 12; and the random seeds taking three possible values (1, 2 and 3).

**Baselines.** In order to test our hypothesis, we test two baselines with incomplete information that effectively hide the context or the information about the metaphorical expression being tested. We will use the following sentence as our running example, with *dark* being the PME of the sentence: *The latest developments move us closer to a dark age.* We test the following three configurations according to the input shown to the model:

1. **Default:** *The latest developments move us closer to a <PME>dark</PME> age.*[3]

2. **Baseline 1 – Only PME**: *dark.*

---

[2]In Appendix A, we include more details on how individual datasets were sampled and preprocessed.

[3]*<PME>* is a special token to indicate the PME position.

|        |              | Def   | PME          | Masked        |
| ------ | ------------ | ----- | ------------ | ------------- |
|        | Trofi        | 75.78 | 56.67 (-25.2%) | 74.42 (-1.8%)  |
| NLP    | TSV_AN       | 87.36 | 52.49 (-39.9%) | 80.88 (-7.4%)  |
|        | TSV_SVO      | 92.33 | 50.27 (-45.5%) | 90.98 (-1.5%)  |
|        | SE2013_ALL   | 86.46 | 49.39 (-42.9%) | 74.60 (-13.7%) |
|        | SE2013_LEX   | 92.92 | 63.46 (-31.7%) | 89.21 (-4.0%)  |
|        | MAD_FEWSHOT  | 94.72 | 89.14 (-5.9%)  | 71.72 (-24.3%) |
| PIE    | MAD_ONESHOT  | 88.21 | 86.24 (-2.2%)  | 65.65 (-25.6%) |
|        | MAD_ZEROSHOT | 81.54 | 77.56 (-4.9%)  | 64.44 (-21.0%) |
|        | PIE          | 88.81 | 91.14 (+2.6%)  | 81.0 (-8.8%)   |
|        | MAGPIE_L     | 94.47 | 90.87 (-3.8%)  | 81.25 (-14.0%) |
|        | MAGPIE_R     | 88.99 | 79.23 (-11.0%) | 80.25 (-9.8%)  |
| VUAC   | VUAC_ST1     | 82.52 | 69.82 (-15.4%) | 70.64 (-14.4%) |
|        | VUAC_ST2     | 82.22 | 73.10 (-11.1%) | 66.48 (-19.1%) |

Table 2: Macro-F1 scores for the original split of existing metaphor identification datasets, under three settings: default setting (*Def*), only PME (*PME*), and with the PME being masked (*Masked*). Relative performance difference of the baselines with respect to the default setting is added in brackets.

3. **Baseline 2 – Masked PME:** *The latest developments move us closer to a <masked> age.*

**Evaluation metrics.** Macro-F1 is chosen as our objective evaluation metric on the validation set during hyper-parameter optimization and is the main metric used in our experiment. Our choice of Macro-F1 was because it gives a synthetic view of the performances of the models for both classes, and datasets with very different label distributions. Accuracy, precision, recall and F1 results for the metaphor class can be found in Appendix B.

## 3.2 Results

The main results are displayed in Table 2. In general, gaps in performance should only be compared within datasets, and are not comparable across datasets because of different sampling techniques. For instance, the balance between metaphorical expressions may be vastly different in different datasets, and the performance gap may also reflect this.

As can be observed, the results of models with incomplete information are extremely competitive, even in the case when only a PME is included. The results also show how different datasets suffer from diverse types of bias. For instance, the *only PME* baseline appears to be insufficient in NLP datasets, with the *masked* baseline being close to the *default* setting. In contrast, in datasets such as MAD, PIE or MAGPIE, the *only PME* baseline is more competitive (with relative drops often lower

than 5%), even surpassing the default upperbound in PIE.

## 3.3 Analysis: Random and Lexical Splits

**Setting.** Similarly to Shwartz et al. (2016), in this experiment we consider datasets with random and lexical splits. A random split is simply a random allocation of instances for the training and test sets. In a lexical split, however, we ensure that a target word (the PME in our case) in the test set is not included in the training set. For a better generalisation, the experiments for this analysis were repeated on five different splits using 5-fold cross-validation.[4] For each fold, 70% of the instances in the training set, 10% in the validation set and 20% in the test set.

**Results.** Table 3 shows the results for the random and lexical splits. The trends observed in these splits are similar to the original splits. This now includes the datasets coming from the psycholinguistic literature, with minimal performance drops of the baseline masking the PME (lower than 5% in all cases). When the natural distribution is not used, the gap between the baselines and the model seeing all the information is in some cases small, with the performance of these baselines being nontrivial. This is not the case in all the datasets, such as MOH, which differs in its construction method from most of the other datasets.

**Random vs lexical splits.** There are cases in which the random split may mislead the model, especially when the number of examples is limited. For example, one dataset may contain only literal examples in the training set for a PME, and the model may wrongly conclude that all instances in the test set of that PME are literal, when this may not be the case. This would not happen in the lexical split. For instance, the DUNN dataset was created with three instances per PME, one literal and two metaphorical. Similarly, for CARD there are two instances per PME, one literal and one metaphorical.

**The case of VUAC.** VUAC has the particularity of following a natural distribution, and has been sampled differently by different researchers to frame it into a binary classification task. Despite its simplicity, our VUAC sampling approach

---

[4] Given their large size (i.e., more than 10,000 test instances), the MAGPIE and VUAC-derived datasets experiments are done on a single train/valid/test split.

|  | Dataset | Random split | | | Lexical split | | |
|  |  | Default | PME | Masked | Default | PME | Masked |
|---|---|---|---|---|---|---|---|
| PyschoLing | CARD_N | 87.38 | 41.21 (-52.8%) | 85.81 (-1.8%) | 89.64 | 49.12 (-45.2%) | 86.72 (-3.3%) |
|  | CARD_V | 83.75 | 41.88 (-50.0%) | 82.60 (-1.4%) | 86.0 | 47.10 (-45.2%) | 82.07 (-4.6%) |
|  | JANK | 82.66 | 38.61 (-53.3%) | 81.23 (-1.7%) | 82.02 | 45.17 (-44.9%) | 80.67 (-1.6%) |
| NLP | TroFi | 88.26 | 70.23 (-20.4%) | 84.33 (-4.5%) | 81.24 | 55.75 (-31.4%) | 77.28 (-4.9%) |
|  | TSV_AN | 89.19 | 59.02 (-33.8%) | 79.03 (-11.4%) | 86.92 | 62.22 (-28.4%) | 77.72 (-10.6%) |
|  | TSV_AN_L2 | 89.19 | 59.02 (-33.8%) | 79.03 (-11.4%) | 87.53 | 59.3 (-32.3%) | 78.13 (-10.7%) |
|  | GUT | 98.25 | 64.98 (-33.9%) | 95.33 (-3.0%) | 95.01 | 43.64 (-54.1%) | 93.96 (-1.1%) |
|  | GUT_L2 | 98.25 | 64.98 (-33.9%) | 95.33 (-3.0%) | 97.54 | 65.56 (-32.8%) | 94.77 (-2.8%) |
|  | MOH | 63.71 | 54.31 (-14.8%) | 53.97 (-15.3%) | 63.86 | 54.43 (-14.8%) | 54.54 (-14.6%) |
|  | LLC | 86.07 | 79.80 (-7.3%) | 73.21 (-14.9%) | 84.49 | 78.54 (-7.0%) | 71.81 (-15.0%) |
|  | CHAK | 64.53 | 70.73 (+9.6%) | 39.98 (-38.0%) | 59.85 | 70.07 (+17.1%) | 38.81 (-35.2%) |
|  | NEU | 71.42 | 48.54 (-32.0%) | 65.29 (-8.6%) | 73.51 | 32.46 (-55.8%) | 74.60 (+1.5%) |
|  | DUNN | 66.08 | 38.68 (-41.5%) | 61.4 (-7.1%) | 54.47 | 39.81 (-26.9%) | 57.40 (+5.4%) |
| PIE | IDIX | 93.79 | 85.24 (-9.1%) | 84.52 (-9.9%) | 75.29 | 62.38 (-17.1%) | 73.81 (-2.0%) |
|  | PVC_V | 84.46 | 73.39 (-13.1%) | 78.05 (-7.6%) | 63.31 | 47.56 (-24.9%) | 59.3 (-6.3%) |
|  | VNC | 93.94 | 89.96 (-4.2%) | 79.12 (-15.8%) | 76.59 | 63.27 (-17.4%) | 69.09 (-9.8%) |
|  | SE2013_ALL | 91.17 | 82.37 (-9.7%) | 84.51 (-7.3%) | 77.04 | 46.63 (-39.5%) | 76.68 (-0.5%) |
|  | SE2013_LEX | 92.54 | 61.16 (-33.9%) | 89.29 (-3.5%) | 80.39 | 43.27 (-46.2%) | 77.58 (-3.5%) |
|  | MAD | 94.58 | 86.99 (-8.0%) | 75.89 (-19.8%) | 77.38 | 70.14 (-9.4%) | 68.76 (-11.1%) |
|  | PIE | 94.86 | 94.34 (-0.5%) | 86.20 (-9.1%) | 87.1 | 87.5 (+0.5%) | 74.45 (-14.5%) |
|  | MAGPIE | 94.83 | 91.07 (-4.0%) | 81.17 (-14.4%) | 87.66 | 79.85 (-8.9%) | 78.38 (-10.6%) |
| VUAC | VUAC_DO | 75.46 | 77.35 (+2.5%) | 62.84 (-16.7%) | 74.02 | 74.43 (+0.6%) | 58.55 (-20.9%) |
|  | VUAC_ST1 | 82.81 | 70.19 (-15.2%) | 70.93 (-14.4%) | 73.81 | 62.28 (-15.6%) | 67.88 (-8.0%) |
|  | VUAC_ST2 | 85.13 | 75.27 (-11.6%) | 68.22 (-19.9%) | 78.53 | 66.70 (-15.1%) | 69.26 (-11.8%) |
|  | VUAC_BO | 85.42 | 63.26 (-25.9%) | 75.94 (-11.1%) | 82.0 | 64.50 (-21.3%) | 72.42 (-11.7%) |

Table 3: Macro-F1 results, averaged over 5 cross-validation folds, for the default, *only PME* (PME), and *Masked* settings on the random and lexical splits of metaphor identification datasets. The values in parentheses represent the relative performance gap compared to the Default configuration.

(i.e., VUAC_BO) appears to be the generally robust, with over 10 points of difference between the results of the default configuration and the best baseline, both for the random and lexical splits.

## 4 Conclusion

Getting inspiration from previous studies analysing artifacts and biases from NLP datasets, we delved into the field of metaphor identification. By proposing baselines with incomplete information that hide the PME or the context, we show how the performance achieved by a supervised model is closed to the model that uses complete information. This highlights the type of bias that a model is picking up at training time, which differ from what a well-trained model would be expected to learn. Finally, we show that this problem is generally observed in datasets that are artificially constructed, as carefully sampling from datasets (VUAC_BO) stemming from a fully-annotated corpus alleviates this issue.

## Limitations

Since this is a preliminary study on the study of biases in metaphor identification datasets, this comes with its own limitations that could be addressed in future work. For instance, we evaluate the models in a set of pre-defined splits that may include their own biases. We attempt at mitigating this by also including cross-validation settings on lexical and random splits, but this may not provide the full picture. For example, we do not study the effect of context duplication in this work (more details in the last point of Appendix A). In terms of the models evaluated, given computational limitations, we focus on a single language model. Our goal is not to achieve the best possible results, but it is likely that some of the conclusions may slightly differ for different supervised models, including other language models.

A human evaluation in the three settings would be interesting to better understand the inherent biases of the models due to the probability of each PIE to be used metaphorically or the probability of

a context to be paired with a metaphor from dataset artifacts. This could help provide a more complete picture of the reasons behind our findings.

## Ethics statement

Our analysis sheds light on the possible deficiencies of supervised settings and the spurious correlations that supervised models can learn from if the datasets are not carefully designed. As such, researchers should be careful in the claims we can make about such models, in particularly in relation to metaphor identification as we show in this paper. In fact, that there may be other artifacts and biases not considered in this paper.

## Acknowledgments

Jose Camacho-Collados is supported by a UKRI Future Leaders Fellowship. We thank all the authors of the datasets used in this paper for kindly sharing them with us, including those that were not currently available online.

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

## A  Dataset Preprocessing and Sampling

In the following we list a few individual dataset preprocessing and sampling details not included in the main paper.

1. A few instances of the TSV test set were found in the training set, we deduplicate them in the random and lexical splits of the dataset.

2. The original 200 instances of the TSV_AN test set are provided within a full sentence context, which we ignore in our experiments because the training set only contains adjective-noun pairs and finetuned language models generally perform better when the test set is similar to the training set.

3. In the lexical splits columns of Table 3, TSV_AN and TSV_AN_L2 are two lexical splits of the TSV_AN instances respectively on the adjective and the noun. The same reading applies to the GUT dataset. The results for a single random split are shown duplicated in the table, for the two rows.

4. Some sentences of the WSJ appear several times in the TroFi dataset original version, we also deduplicate them in the random and lexical spits of the dataset.

5. The MAGPIE has an original lexical and random split, for which the results are presented in Table 1 (MAGPIE_L and MAGPIE_R). Different random and lexical splits appear in Table 3.

6. In Table 1, the statistics of the original MAD_FEWSHOT dataset are shown. Two other versions with less instances have been shared in Tayyar Madabushi et al. (2022): MAD_ONESHOT and MAD_ZEROSHOT (equivalent to a lexical split). The results for the three original splits appear in Table 2. Our lexical and random splits of the MAD dataset are made from the instances of the few-shot version, that contains the largest number of instances among the three.

7. The PVC dataset contains too few distinct V-PREP instance for a relevant lexical split based on the two words, the split is made on the verb for each PME.

8. A few instances of the IDIX and VNC datasets contain minimal meaningful contextual information in the *only-PME* setting because they are multiword expressions of non consecutive words. For example, the idiom *to rock the boat* appears in the sentence as rock the political boat. Informative context inserted between the terms of a PIE occurs very rarely in those two datasets. Therefore, while they could have an effect in the final results, we decided not to not modify them and simply extracted or tagged the expressions by selecting the string starting and finishing with the PIE.

9. **Context duplication**: The CHAK dataset has the specificity of being made of triples, where nearly identical sentences appear three times. The sentences of a triple have the same context and three different substituted verbal PMEs, among which exactly one is literal and two are figurative. The MOH dataset, constructed from WordNet glosses, also contains several instances with identical context, and alternated verbal PMEs that are near synonyms of each other. Finally, multiple instances of the VUAC datasets may be constructed from the same sentence during the sampling procedure, because several words of each sentence may have been labeled as metaphoric during the original annotation process.

## B Additional evaluation metrics

Tables 4 and 5 provide additional metrics for the random and lexical split analyses, respectively (see Section 3.3). In particular, the tables include accuracy and precision, recall and F1 on the metaphor class. Moreover, for completeness we have included the accuracy results of a naive baseline that outputs the metaphor for all instances.

| | | Accuracy | | | | Precision | | | Recall | | | F1 | | |
|---|---|---|---|---|---|---|---|---|---|---|---|---|---|---|
| | Dataset | Maj | Def | PME | Mask | Def | PME | Mask | Def | PME | Mask | Def | PME | Mask |
| Psy | CARD_N | 50.0 | 87.5 | 44.5 | 85.9 | 90.5 | 42.8 | 86.7 | 83.9 | 33.9 | 85.1 | 87.0 | 35.4 | 85.6 |
| | CARD_V | 50.0 | 83.9 | 42.9 | 82.9 | 87.2 | 40.8 | 83.1 | 80.8 | 36.9 | 83.4 | 83.3 | 38.3 | 83.0 |
| | JANK | 66.7 | 85.3 | 51.1 | 84.2 | 81.2 | 45.1 | 82.2 | 75.0 | 28.4 | 68.5 | 76.5 | 45.1 | 74.2 |
| NLP | TroFi | 57.6 | 88.5 | 71.3 | 84.6 | 91.2 | 73.7 | 88.1 | 88.7 | 78.2 | 84.8 | 89.9 | 75.8 | 86.4 |
| | TSV_AN | 50.3 | 89.3 | 59.4 | 79.2 | 91.3 | 62.2 | 80.2 | 86.9 | 52.0 | 78.0 | 89.1 | 55.8 | 78.9 |
| | GUT | 53.6 | 98.3 | 65.4 | 95.4 | 98.9 | 66.8 | 96.0 | 97.9 | 70.6 | 95.3 | 98.4 | 68.6 | 95.7 |
| | MOH | 78.8 | 78.3 | 73.2 | 73.8 | 48.8 | 32.7 | 33.2 | 35.8 | 22.1 | 20.0 | 40.7 | 45.1 | 23.9 |
| | LLC | 58.7 | 86.5 | 80.5 | 74.1 | 83.1 | 76.8 | 69.2 | 84.5 | 75.8 | 68.1 | 83.8 | 76.2 | 68.4 |
| | CHAK | 66.7 | 69.7 | 74.4 | 66.7 | 76.0 | 80.2 | 66.7 | 79.8 | 81.8 | 100.0 | 77.8 | 81.0 | 80.0 |
| | NEU | 56.0 | 76.0 | 56.0 | 72.0 | 81.3 | 58.8 | 79.9 | 79.8 | 66.0 | 74.5 | 77.6 | 60.4 | 72.9 |
| | DUNN | 66.7 | 71.7 | 63.3 | 71.7 | 78.7 | 66.3 | 73.3 | 77.4 | 96.4 | 93.5 | 76.8 | 77.4 | 81.0 |
| PIE | IDIX | 51.4 | 93.8 | 85.3 | 84.6 | 93.3 | 83.1 | 84.4 | 93.7 | 86.8 | 83.0 | 93.5 | 84.9 | 83.7 |
| | PVC | 65.1 | 85.9 | 76.9 | 80.8 | 89.0 | 79.8 | 83.2 | 89.4 | 86.6 | 88.6 | 89.2 | 82.8 | 85.7 |
| | VNC | 78.5 | 96.0 | 93.5 | 87.0 | 97.1 | 95.0 | 89.8 | 97.9 | 96.8 | 94.1 | 97.5 | 95.9 | 91.9 |
| | SE2013_ALL | 59.5 | 91.5 | 83.2 | 85.3 | 93.5 | 84.5 | 86.3 | 92.3 | 87.9 | 89.3 | 92.8 | 86.2 | 87.8 |
| | SE2013_LEX | 50.6 | 92.6 | 61.9 | 89.3 | 93.4 | 60.4 | 89.0 | 91.9 | 73.7 | 90.2 | 92.6 | 65.8 | 89.6 |
| | MAD | 52.0 | 94.6 | 87.0 | 76.0 | 93.1 | 82.2 | 75.4 | 95.9 | 93.1 | 74.6 | 94.4 | 87.3 | 74.9 |
| | PIE | 52.6 | 94.9 | 94.4 | 86.2 | 93.3 | 92.5 | 83.1 | 96.1 | 95.8 | 89.1 | 94.7 | 94.1 | 86.0 |
| | MAGPIE | 74.7 | 96.1 | 93.3 | 86.7 | 97.4 | 95.4 | 88.6 | 97.3 | 95.5 | 94.3 | 97.4 | 95.5 | 91.4 |
| VUAC | VUAC_DO | 50.4 | 75.5 | 77.4 | 63.0 | 74.1 | 77.1 | 62.1 | 79.1 | 78.4 | 68.0 | 76.5 | 77.7 | 64.9 |
| | VUAC_ST1 | 71.6 | 86.2 | 75.8 | 77.2 | 77.3 | 57.5 | 61.4 | 73.1 | 56.9 | 53.9 | 75.1 | 57.2 | 57.4 |
| | VUAC_ST2 | 84.3 | 92.3 | 86.5 | 84.7 | 76.7 | 56.4 | 51.4 | 73.0 | 61.1 | 39.9 | 74.8 | 58.6 | 45.0 |
| | VUAC_BO | 50.5 | 85.4 | 63.3 | 76.0 | 84.6 | 63.4 | 75.4 | 86.9 | 64.6 | 77.8 | 85.8 | 64.0 | 76.6 |

Table 4: Majority class accuracy (Maj) accuracy is shown in the first result column. Accuracy, precision, recall and F1 results for the metaphor class, averaged over 5 cross-validation folds, for the Default (Def), only PME (PME), and Masked settings on the random splits of metaphor identification datasets appear in the following columns.

| | | Accuracy | | | | Precision | | | Recall | | | F1 | | |
|---|---|---|---|---|---|---|---|---|---|---|---|---|---|---|
| | Dataset | Maj | Def | PME | Mask | Def | PME | Mask | Def | PME | Mask | Def | PME | Mask |
| Psy | CARD_N | 50.0 | 89.7 | 50.0 | 86.7 | 90.1 | 50.0 | 88.6 | 89.5 | 59.4 | 84.4 | 89.7 | 54.0 | 86.4 |
| | CARD_V | 50.0 | 86.1 | 50.0 | 82.1 | 87.6 | 50.0 | 85.4 | 84.3 | 54.3 | 77.9 | 85.6 | 48.7 | 81.2 |
| | JANK | 66.7 | 84.2 | 51.4 | 83.3 | 78.6 | 26.7 | 78.8 | 74.2 | 45.8 | 70.0 | 75.9 | 33.3 | 73.6 |
| NLP | TroFi | 57.6 | 82.3 | 62.9 | 78.5 | 85.1 | 63.4 | 81.2 | 85.0 | 84.1 | 82.9 | 84.7 | 72.2 | 81.6 |
| | TSV_AN | 50.3 | 87.0 | 63.5 | 77.8 | 88.0 | 64.6 | 76.9 | 85.2 | 59.9 | 79.2 | 86.5 | 61.2 | 77.9 |
| | TSV_AN_L2 | 50.3 | 87.6 | 59.5 | 78.2 | 91.7 | 61.5 | 79.6 | 83.0 | 53.4 | 77.0 | 87.1 | 56.8 | 78.0 |
| | GUT | 53.6 | 95.3 | 52.1 | 94.2 | 96.0 | 47.6 | 93.9 | 94.8 | 55.0 | 94.9 | 95.3 | 45.6 | 94.3 |
| | GUT_L2 | 53.6 | 97.6 | 66.0 | 94.8 | 97.8 | 68.1 | 95.8 | 97.7 | 69.6 | 94.5 | 97.7 | 68.5 | 95.1 |
| | MOH | 78.8 | 79.0 | 74.4 | 74.2 | 56.9 | 34.9 | 34.4 | 34.4 | 22.2 | 45.1 | 40.7 | 24.5 | 45.1 |
| | LLC | 58.7 | 85.1 | 79.1 | 72.8 | 83.2 | 73.7 | 67.5 | 80.0 | 76.7 | 66.0 | 81.6 | 75.2 | 66.7 |
| | CHAK | 66.7 | 64.3 | 73.2 | 63.7 | 74.6 | 81.0 | 65.6 | 71.1 | 77.9 | 95.3 | 72.2 | 79.3 | 77.6 |
| | NEU | 56.0 | 76.0 | 56.0 | 77.0 | 84.4 | 43.0 | 82.2 | 73.6 | 80.0 | 81.0 | 73.8 | 54.5 | 79.1 |
| | DUNN | 66.7 | 66.7 | 56.7 | 70.0 | 72.7 | 64.8 | 74.7 | 82.5 | 75.0 | 87.5 | 76.2 | 65.9 | 79.8 |
| PIE | IDIX | 51.4 | 75.5 | 63.2 | 74.1 | 76.5 | 66.3 | 74.5 | 77.4 | 62.5 | 74.9 | 75.1 | 62.5 | 73.5 |
| | PVC_V | 65.1 | 69.5 | 59.2 | 67.3 | 73.4 | 66.8 | 68.9 | 78.9 | 76.3 | 80.0 | 75.3 | 67.4 | 72.8 |
| | VNC | 78.5 | 84.3 | 73.4 | 81.8 | 90.4 | 85.6 | 86.4 | 90.5 | 80.6 | 91.2 | 89.5 | 81.7 | 88.5 |
| | SE2013_ALL | 59.5 | 79.2 | 49.2 | 79.1 | 81.5 | 57.8 | 78.7 | 82.3 | 50.6 | 85.2 | 79.9 | 48.7 | 80.4 |
| | SE2013_LEX | 50.6 | 81.3 | 47.1 | 78.7 | 80.2 | 41.0 | 79.8 | 86.0 | 50.9 | 78.2 | 81.8 | 42.1 | 78.1 |
| | MAD | 52.0 | 78.1 | 71.1 | 69.3 | 78.5 | 65.9 | 68.0 | 75.0 | 79.8 | 66.1 | 76.0 | 72.0 | 66.9 |
| | PIE | 52.6 | 87.2 | 87.6 | 74.7 | 84.0 | 82.4 | 71.4 | 90.3 | 94.1 | 79.1 | 86.8 | 87.8 | 74.9 |
| | MAGPIE | 73.7 | 90.2 | 84.9 | 83.4 | 94.4 | 88.4 | 88.4 | 92.2 | 91.5 | 89.2 | 93.3 | 89.9 | 88.8 |
| VUAC | VUAC_DO | 57.6 | 74.5 | 75.0 | 59.6 | 78.5 | 78.7 | 64.7 | 76.7 | 77.5 | 65.7 | 77.6 | 78.0 | 65.2 |
| | VUAC_ST1 | 68.5 | 77.0 | 67.0 | 73.2 | 62.7 | 47.8 | 58.5 | 66.8 | 49.9 | 51.6 | 64.7 | 48.8 | 54.8 |
| | VUAC_ST2 | 82.6 | 88.3 | 83.3 | 84.2 | 68.8 | 53.2 | 56.1 | 60.0 | 36.3 | 41.7 | 64.1 | 43.2 | 47.9 |
| | VUAC_BO | 53.4 | 82.2 | 65.5 | 73.0 | 80.7 | 64.9 | 71.5 | 87.7 | 77.0 | 82.1 | 84.1 | 77.0 | 76.5 |

Table 5: Majority class accuracy (Maj) accuracy is shown in the first result column. Accuracy, precision, recall and F1 results for the metaphor class, averaged over 5 cross-validation folds, for the Default (Def), only PME (PME), and Masked settings on the lexical splits of metaphor identification datasets appear in the following columns.