# OpenReview forum: "Construction Artifacts in Metaphor Identification Datasets"
_EMNLP/2023/Conference — EMNLP 2023 Main_

### Official Review · Reviewer_JUSd · 2023-08-03

**Soundness:** 3

**Excitement:**

3: Ambivalent: It has merits (e.g., it reports state-of-the-art results, the idea is nice), but there are key weaknesses (e.g., it describes incremental work), and it can significantly benefit from another round of revision. However, I won't object to accepting it if my co-reviewers champion it.

**Paper Topic And Main Contributions:**

This paper designed multiple experiments and concluded that existing metaphor identification datasets can be used by fully ignoring the context of target words.  The authors also experimented to show the influence of the sampling method on multiple datasets.

**Reasons To Accept:**

The idea is interesting. The authors did many experiments to evaluate their thoughts.

**Reasons To Reject:**

1. The conclusion and analysis in this paper was very strong. The authors used words like “extremely competitive” in line 233 to explain the results in Table 2. But only 2 datasets have higher performance in PME/Masked than the default setting. How could it become a general conclusion? A deeper analysis is expected in the discussion part.
2. The same issue shows in Table 3. Half datasets show higher performance in random/lexical split than the default setting. The conclusion is for all the datasets. I don't think the results are clear enough to come to such a strong conclusion.
3. The sampling method was only evaluated on VUA. Whether it could be generalized on other datasets was not evaluated.

**Reproducibility:**

4: Could mostly reproduce the results, but there may be some variation because of sample variance or minor variations in their interpretation of the protocol or method.

**Reviewer Confidence:**

4: Quite sure. I tried to check the important points carefully. It's unlikely, though conceivable, that I missed something that should affect my ratings.

**Typos Grammar Style And Presentation Improvements:**

Line 151, "in two different tracks", what two tracks?

Line 152, what is NN for?

---

> ### Author Rebuttal · Authors · 2023-08-28
>
> We thank the reviewer for reviewing our paper. By reading the review, we believe there has been a general misunderstanding with respect to the main objectives and claims of the paper. In this paper we are not interested in testing whether the baselines with incomplete information are better than the default model with complete information. In some of the results, the baselines are indeed better than the default model, but these are just rare cases as the reviewer points out. The claim of the paper is that these baselines, given that they lack the complete information, obtain unreasonably high results in most datasets. As we would expect, these results are still worse than the default configuration with complete information. We present our detailed response for each comment below.
>
>
> __The conclusion and analysis in this paper was very strong. The authors used words like “extremely competitive” in line 233 to explain the results in Table 2. But only 2 datasets have higher performance in PME/Masked than the default setting. How could it become a general conclusion? A deeper analysis is expected in the discussion part.__
>
> We agree that “extremely competitive” may be a bit misleading. Indeed, our claim is not meant to imply that the baselines are better than the default model, which is not the case. In the context of our paper, especially in the case of the PME alone with no context information, the model performance should be theoretically bounded by the probability of each PME to be used metaphorically. This is definitely not the case given the high results of the baselines. Therefore, our results suggest that the artifacts in dataset construction make the tasks artificially simpler than they would be in a more natural context.  We will add this discussion when talking about the “competitiveness” of the baselines.
>
> __The same issue shows in Table 3. Half datasets show higher performance in random/lexical split than the default setting. The conclusion is for all the datasets. I don't think the results are clear enough to come to such a strong conclusion.__
>
> Please refer to the previous question. Again, our claim is not that the baselines are better than the default model with full information, but rather that the baselines are better than what we could expect without relying on complete information. We will clarify this.
>
> __The sampling method was only evaluated on VUA. Whether it could be generalised on other datasets was not evaluated.__
>
> The sampling method could not really be used on other existing datasets. The VUA is the only project where every word in a corpus has been labeled as metaphorical or literal. The other datasets only label one word in a given context, with sentences extracted from different corpora and, in some cases, generated ad-hoc.

---

### Official Review · Reviewer_CchZ · 2023-08-04

**Soundness:** 4

**Excitement:**

4: Strong: This paper deepens the understanding of some phenomenon or lowers the barriers to an existing research direction.

**Paper Topic And Main Contributions:**

This paper investigates biases in metaphorical identification datasets. Two baselines are used, one masking the metaphorical expression and one using only the metaphorical expression. Both baselines are compared with a default model, using the text along with information about the position of the metaphorical expression. The findings show that different datasets suffer from different biases, and in some cases the baselines outperform the default setting. The authors experiment also with two splits, a random one and one where train and validation subsets do not share the same metaphorical spans.

**Questions For The Authors:**

A. The lexical split (PME in test but not train) is more challenging, as the other way round can also be considered as data leaking. However, the results with the default model are better compared to the random split in some cases (e.g., Table 3, DUNN). Why?

B.  As in the case of Trofi, the baseline has outperformed the default model. Can you please provide examples from the respective datasets that could indicate why?


**Reasons To Accept:**

- Benchmark of several dataset versions, serving as a good reference point
- Revealing biases that directly affect the future usage of the studied datasets
- Well defined hypothesis and method: employing BERT as a building block, to assess the hypothesis by ablating the input

**Reasons To Reject:**

- Examples are not shared to allow the reader to better understand the findings, but this can easily be addressed in the camera ready.
- Statistical significance is missing, putting reproducibility of the results at stake. I agree that the original dataset train/test splits should be respected, but one does not exclude the other (e.g., two more splits could be used, so that averages are reported in the appendix).

**Reproducibility:**

4: Could mostly reproduce the results, but there may be some variation because of sample variance or minor variations in their interpretation of the protocol or method.

**Reviewer Confidence:**

3: Pretty sure, but there's a chance I missed something. Although I have a good feel for this area in general, I did not carefully check the paper's details, e.g., the math, experimental design, or novelty.

**Typos Grammar Style And Presentation Improvements:**

- In 2.2, in sentence context, both future and past tense are used.
- In 2.3, please consider explaining what a PIE is and how it is different from a PME.
- In L282-283, please clarify the datasets you are referring to, in order to assist the reader.

---

> ### Author Rebuttal · Authors · 2023-08-28
>
> We thank the reviewer for their supportive comments and carefully reviewing our paper.
>
> __A. The lexical split (PME in test but not train) is more challenging, as the other way round can also be considered as data leaking. However, the results with the default model are better compared to the random split in some cases (e.g., Table 3, DUNN). Why?__
>
> There are cases in which the random split may mislead the model, especially when the number of examples is limited. For example, one dataset may contain only literal examples in the training set for a PME, and the model may wrongly conclude that all instances in the test set of that PME are literal, when this may not be the case. This wouldn’t happen in the lexical split. For instance, the DUNN dataset was created with three instances per PME, one literal and two metaphorical. Similarly, for CARD there are two instances per PME, one literal and one metaphorical.
>
> Please find below an example of a random split in DUNN:
>
> - Possible random split:
>
>     - Training data :
>         - It was essential to dress according to your status and to be able to decipher other people's attire. (Met..)
>         - More and more restaurants are energetically finding ways to help customers decipher their wine lists. (Met.)
>         - …
>
>     - Test data :
>         - It was only 100 years ago that people were beginning to decipher the Rosetta Stone. (Lit.)
>         - …
>
> Finally, please note that the DUNN dataset is the smallest dataset considered, with only 60 instances in total. As such, we even hesitated to include it in the table as some variations may be due to the small size.
>
> __B. As in the case of Trofi, the baseline has outperformed the default model. Can you please provide examples from the respective datasets that could indicate why?-__
>
> We thank the reviewer for noticing this odd number. The mask setting never outperforms the baseline for the TroFi dataset in none of the splits. The correct value for the mask baseline in Table 2 should be 74.42, slightly inferior to the default setting, 75.78. We triple checked manually all the scores in Table 2 and Table 3 and it is the only error in the tables. In general, however, we consider baselines scores that slightly outperform the default models to be anecdotal evidence of how context is not required (and in these cases it is not even helpful) for models to perform well on these tasks. There are no real reasons why these baselines would outperform the default model, but add to the main hypothesis of how models that rely entirely on artificial cues can perform almost as well as models with complete information.
>
> __Additional comments in the reasons to reject :__
>
> __Statistical significance is missing, putting reproducibility of the results at stake. I agree that the original dataset train/test splits should be respected, but one does not exclude the other (e.g., two more splits could be used, so that averages are reported in the appendix).__
>
> Thank you for these suggestions, we will add statistical significance results in the camera-ready version of the paper. In general, the vast majority of the results from the baselines are significantly different statistically (better in this case) than the random and most frequent class baselines (we will add these tests in the final version of the paper). As for the splits, we indeed respected the original splits as there are different reasons for why these were given. Nonetheless, we agree that more splits can be used for the lexical and random split analysis, which we will add by reporting the average of the results and standard deviation, including the detailed results for each split in the appendix.

---

### Official Review · Reviewer_PbYa · 2023-08-06

**Soundness:** 3

**Excitement:**

3: Ambivalent: It has merits (e.g., it reports state-of-the-art results, the idea is nice), but there are key weaknesses (e.g., it describes incremental work), and it can significantly benefit from another round of revision. However, I won't object to accepting it if my co-reviewers champion it.

**Paper Topic And Main Contributions:**

This paper focuses on the task of metaphor detection in a setting in which a single word PME (potentially metaphorical entity) alongside with the context it occurs in is provided to the system and the task is judge whether the usage is metaphorical or not. This study shows that simple BERT-base models for metaphor detection across a variety of datasets perform nearly as well as in the case when the PME is not provided and only the context is provided. Moreover, when PME alone is provided, the performance is non-trivially competent although much worse than the full and the context-only systems. The paper also claims that when considering datasets that are constructed from natural distributions over text rather than artificial construction of metaphor datasets, the performance gap between full systems and context-only/PME-only systems is greater indicating that most metaphor detection datasets might suffer from spurious correlation issue in which PME or context alone is sufficient to determine metaphoricity.

**Reasons To Accept:**

– The paper considers a wide variety of datasets and consistently shows that systems mostly can perform rather well with the knowledge of context alone compared to having access to the full sentence. This finding is interesting and might spur discussion on task specification/dataset creation for analyzing metaphoricity.

**Reasons To Reject:**

– The second claim about greater gap when more natural distribution over text is used for datasets (VUAC) doesn’t seem to be well supported in Table 3. There are plenty of rows showing  greater gaps in other datasets.

– Limited base rate evaluation: While the identified problem of models not using PME or context for making prediction seems serious, it needs to be compared against a more natural human baseline. For example, given just the context, it might be sufficient for humans as well to determine the metaphoricity due to prior knowledge and language use without needing to know the identity of the PME. If humans perform as well as these systems, then it would indicate the issue doesn’t exist with the datasets but rather with the task specification. Similarly, certain words could show a greater natural affinity to be PMEs – this affinity should be compared against the systems performance to judge whether they overestimate the reliance on PME/Context alone to judge metaphoricity.


**Reproducibility:**

4: Could mostly reproduce the results, but there may be some variation because of sample variance or minor variations in their interpretation of the protocol or method.

**Reviewer Confidence:**

5: Positive that my evaluation is correct. I read the paper very carefully and I am very familiar with related work.

---

> ### Author Rebuttal · Authors · 2023-08-28
>
> We thank the reviewer for carefully reading our manuscript and providing insightful comments
>
>
> __The second claim about greater gap when more natural distribution over text is used for datasets (VUAC) doesn’t seem to be well supported in Table 3. There are plenty of rows showing greater gaps in other datasets.__
>
> Thank you for the comment, as our claim may be indeed misleading, and we should better explain this. In general, gaps in performance should only be compared within datasets, and are not comparable across datasets because of different sampling techniques. For instance, the balance between metaphorical expressions may be vastly different in different datasets, and the performance gap may also reflect this.The VUAC has the particularity of following a natural distribution, and has been sampled differently by different researchers to frame it into a binary classification task.The point that we wanted to convey here is that when the natural distribution is not used, the gap between the baselines and the model seeing all the information is in some cases small, with the performance of these baselines being non-trivial. This is not the case in all the datasets, such as MOH, which differs in its construction method from most of the other datasets since it relies on WordNet glosses and identified metaphorical senses of verbs. We will clarify this in the paper.
>
> __Limited base rate evaluation: While the identified problem of models not using PME or context for making prediction seems serious, it needs to be compared against a more natural human baseline. For example, given just the context, it might be sufficient for humans as well to determine the metaphoricity due to prior knowledge and language use without needing to know the identity of the PME. If humans perform as well as these systems, then it would indicate the issue doesn’t exist with the datasets but rather with the task specification. Similarly, certain words could show a greater natural affinity to be PMEs – this affinity should be compared against the systems performance to judge whether they overestimate the reliance on PME/Context alone to judge metaphoricity.__
>
> Thank you for this remark, we fully agree with the two points raised and agree that a human evaluation would be interesting to fully understand the reasons behind our findings. We deemed this analysis out of the scope of this short paper, but we will take this suggestion onboard for a comment in the future work section.

---

### Meta-Review · Area_Chair_EwD8 · 2023-09-18

**Recommendation:** 4

**Metareview:**

This work analyzes how existing metaphor identification datasets can be gamed by fully ignoring the context in which the potential metaphorical expression occurs.

The reviewers ratings range from 3-4 with respect to soundness and excitement, and raised several critical issues which the authors tackle in their rebuttal and generally commit themselves to include in the final version (e.g. claim clarification, adding statistical analyses, etc.). As one reviewer notes, the work lacks a human baseline which would greatly improve the work. This, however, implies great additional work and the authors deemed this out of the scope of the short paper.

---

### Decision · Program_Chairs · 2023-10-07

**Decision:**

Accept-Main

**Comment:**

This work analyzes how existing metaphor identification datasets can be gamed by fully ignoring the context in which the potential metaphorical expression occurs.

The reviewers ratings range from 3-4 with respect to soundness and excitement, and raised several critical issues which the authors tackle in their rebuttal and generally commit themselves to include in the final version (e.g. claim clarification, adding statistical analyses, etc.). As one reviewer notes, the work lacks a human baseline which would greatly improve the work. This, however, implies great additional work and the authors deemed this out of the scope of the short paper.